# Utility of Diagnostic Imaging in the Early Detection and Management of the Fournier Gangrene

**DOI:** 10.3390/diagnostics12102320

**Published:** 2022-09-26

**Authors:** Piotr Sumisławski, Janusz Kołecki, Martyna Piotrowska, Maciej Kotowski, Marcin Szemitko, Jerzy Sieńko

**Affiliations:** 1Department of Neurosurgery, University Hospital Carl Gustav Carus, Technische Universität Dresden, Fetscherstr. 74, 01307 Dresden, Germany; 2Department of General and Dental Radiology, Pomeranian Medical University, Al. Powstańców Wielkopolskich 72, 70-111 Szczecin, Poland; 3Department of General Surgery and Transplantation, Pomeranian Medical University, Al. Powstańców Wielkopolskich 72, 70-111 Szczecin, Poland; 4Department of Interventional Radiology, Pomeranian Medical University, Al. Powstańców Wielkopolskich 72, 70-111 Szczecin, Poland

**Keywords:** Fournier’s gangrene, magnetic resonance imaging, 3D model, necrotizing fasciitis

## Abstract

Fournier gangrene represents a urologic emergency. It is a rapidly progressing necrotizing fasciitis that comprises the perineal, perianal, and genital regions and has a high mortality rate. Diagnosis is usually made clinically, but radiological diagnostics, such as ultrasound (US), computed tomography (CT), or magnetic resonance imaging (MRI), can determine the extent of the disease in relation to pelvic structures. Early and accurate diagnosis precipitates the initiation of the effective treatment and, thus, affects the outcome of the therapy. The article reports an illustrative case study of a patient with Fournier gangrene, secondary to a perianal fistula and perianal abscess with a massive accumulation of fluid around the anus and testicles, requiring unilateral orchidectomy. Rapid radiological diagnosis via MRI enabled precise assessment of the degree of the disease, early surgical intervention, and a successful outcome.

## 1. Introduction

Fournier gangrene is a rapidly extending necrotizing fasciitis of the urogenital and perianal area. It was first described by Baurienne in 1764. The name comes from French venereologist Jean Alfred Fournier, who presented five cases during his lectures in 1883 [1]. The average age of patients ranges from 50 to 60, and the mortality rate is between 20–30% [2]. The incidence rate shows a male-to-female ratio of 10:1. The cause of the disease can be identified in 90% of cases.

The possible sources of FG involve the gastrointestinal tract (30–50%), including perianal abscesses, anal fissures, anal fistula, and colonic perforations and the genitourinary tract (20–40%), such as bulbourethral gland infection, urethral injury, epididymitis, orchitis or lower urinary tract infection, and cutaneous injuries (20%) due to hidradenitis suppurativa, scrotal pressure ulcer, and trauma. The presence of foreign bodies may also lead to the disease. Some conditions that depress the cellular immunity may predispose a patient to the development of Fournier gangrene. The high-risk factors include the following: diabetes mellitus, morbid obesity, alcoholism, immunosuppression, chemotherapy, or chronic corticosteroid use [3,4]. The presence of comorbidities, such as diabetes mellitus, heart disease, renal failure, and kidney diseases, was described as associated with an increased risk of mortality [5].

Necrosis in FG is caused by bacterial inflammation of the vessels and vessel occlusion. Infection extends along the perineal fascia area (superficial perineal fascia and deep perineal fascia) to the scrotum and penis via the Buck’s and dartos fascia or to the abdominal wall via the Scarpa fascia. The testicles are usually not involved as their blood supply is separated from the affected region and comes from the direct abdominal aorta. Testicular involvement might indicate a retroperitoneal source or spread of the infection [3].

FG is a very insidious disease as almost 40% of patients do not present any symptoms [4]. The initial clinical presentation typically involves genital and/or perianal pain along with tenderness, usually with oedema of the overlying skin, pruritus, crepitus, and fever [3,4]. Depending on the source of the infection, symptoms can vary and begin with the signs of diseases related to FG, such as pain in the perianal area with constipation caused by perianal abscess, which can spread downwards resulting in FG [3,6]. Fascial destruction extends rapidly with the rate of up to 2–3 cm per hour [7]. This fulminant necrosis may lead to systemic complications, such as sepsis, multiple organ failure, septic shock, and even death [5].

Because of the life-threatening condition, it requires quick and accurate diagnosis. Several different scores, such as the Laboratory Risk Indicator for Necrotizing Fasciitis (LRINEC), Fournier Gangrene Severity Index (FGSI), and Uludag Fournier Gangrene Severity Index (UFGSI), were established to predict the risk of FG, the severity of the disease, and the outcome [8,9,10]. The diagnosis is usually made clinically, but radiological imaging—ultrasound (US), computed tomography (CT), magnetic resonance imaging (MRI)—can be very useful in clinically ambiguous and complex cases [2,11]. CT is a modality of choice in diagnostic imaging of FG, but there are cases where MRI should be preferred, especially in cases of FG combined with perianal fistula and abscess, because of its higher soft-tissue resolution and detailed assessment of the disease’s extent. US can be considered in the cases of patients who cannot be transported for CT/MRI scanning [6,12,13]. Therapy involves administration of broad-spectrum antibiotics and aggressive surgical debridement of necrotic tissues [14]. Several new promising therapies, such as hyperbaric oxygenation (HBO), vacuum-assisted wound closure (VAC), and maggot therapy (blowfly larvae), were also reported [15].

## 2. Methods

### 2.1. Vital and Laboratory Parameters

To assess the probability of necrotizing fasciitis and prognosis using the Laboratory Risk Indicator for Necrotizing Fasciitis (LRINEC) and the Fournier’s Gangrene Severity Index (FGSI), the following laboratory biomarkers were determined: hematocrit, hemoglobin, leukocytes, serum creatinine, sodium, potassium, glucose, bicarbonate, and CRP, along with vital parameters, including body temperature, heart rate, and respiratory rate.

### 2.2. Radiologic Imaging

MRI was performed with a 1.5-T machine GE Signa, HDxt, General Electric Medical Systems with an 8-channel body coil. The following images were obtained: T2-weighted FRFSE images without fat saturation (TR 6960 ms, TE 100,5 ms, 5-mm slice thickness, 1.0-mm gap) and with fat saturation (TR 7420 ms, TE 98,3 ms, 5-mm slice thickness, 1.0-mm gap); T1-weighted FSE images without the contrast agent (TR 660 ms, TE 9,3 ms, 5-mm slice thickness, 1.0-mm gap) and then after injecting 20 mL of gadodiamide (Omniscan) in the dynamic LAVA gradient echo pulse sequence (TR 4,2 ms, TE 2.0 ms, 4.4-mm slice thickness with 2.2-mm overlap) as well as T1-weighted FSE with FAT SAT (TR 2240 ms, TE 9,3 ms, 5-mm slice thickness, 1.0-mm gap).

## 3. Case Presentation

A 59-year-old obese (BMI 31.5 kg/m^2^) Caucasian male with a past medical history of hypertension and ischemic heart disease was admitted to the emergency department with complaints of rectal pain radiating to the scrotum, which started 10 days prior to presentation, urinary retention lasting 24 h, and diarrhea for 11 days. There was no history of diabetes mellitus or alcohol abuse.

Vital signs (blood pressure, heart rate, respiration rate) were stable. A physical examination revealed erythema and induration in the area of the sacral region, indicating a perianal abscess, along with hemorrhoids in the rectal examination. Laboratory results revealed leukocytosis (23.650/μL) with neutrophilia (19.830/μL, 83%) and an elevated C-reactive protein level (173.8 mg/dL). Biochemical parameters, such as serum creatinine, sodium, potassium, bicarbonate, and glucose, along with hemoglobin and hematocrit, were within the reference range. On admission, a perianal abscess incision and drainage were performed. To assess the extent of the disease, magnetic resonance imaging (MRI) was performed.

MRI revealed a break in the continuity of the skin on the left side of the posterior anal wall with a width of 5 mm and high signal intensity on T2-weighted FSE sequences, 35 mm from the anal sphincter. It connected the anal canal with perianal fluid collection adjacent to the posterior anal wall with dimensions of approximately 15 × 20 × 20 mm (Figure 1). Two blisters, with fluids spread along the lateral anal walls with a diameter of 5–10 mm on the right side (Figure 2) and 10–15 mm on the left side, converged on the anterior wall (Figure 3 and Figure 4). An abnormal signal intensity extended downwards to the buttocks on both sides of the anus (Figure 5), but only the left side showed inflammatory infiltration with numerous gas foci in the area of 60 mm and extending to the perineum (Figure 6).

After the MRI scan and diagnosis of transsphincteric perianal fistula, perianal abscess, and Fournier’s gangrene, the patient was taken up for an emergency scrotal andperianal debridement followed by a left orchiectomy with bilateral drainage of the perianal area (Figure 7). The patient had dual intravenous antibiotic therapy (vancomycin—1g twice daily, piperacillin/tazobactam—4,5g four times daily) until the positive culture results were obtained. The microbiological culture revealed Escherichia coli, Enterococcus faecalis, and Bacteroides stercoris sensitive to piperacillin/tazobactam. The piperacillin/tazobactam antibiotic was continued. Wound dressing was performed every second day and reconstructed by secondary suturing on the 20th postoperative day with a good cosmetic outcome (Figure 8). The patient was discharged home 23 days after presentation with an excellent prognosis. A unique form and extension of fluid collections as a 3D model was designed and visualized using previously described software 3D Slicer v4.8.1 software [16,17] and was based on the MRI scan (Appendix A).

## 4. Discussion

Fournier gangrene, a rapidly progressing disease with a potentially fatal outcome, requires increased attention in the diagnostic approach. Due to many possible sources of infection, it may have various clinical presentations, including an indolent course that may hinder the final diagnosis [3,4]. The intensity of the symptoms may be inadequate to physical findings, which indicate the importance of radiological imaging in early diagnosis, assessment of the extension of the disease, and rapid induction of effective treatment [4]. In our case, a clinical examination showed an abscess in the perianal area with increased inflammation markers but without any further symptoms or abnormal laboratory findings. After the initial incision and drainage of the perianal abscess, an MRI was performed to broaden the diagnosis. It revealed a massive spread of fluid collections and Fournier gangrene of the gluteal muscles as well as the perineal and scrotal area. More aggressive therapy was administered and resulted in a successful outcome. CT and US are usually the methods of choice performed with FG as they are widely available and serve as effective tools for the detection of soft tissue thickening or gas [11,13,18]. However, MRI remains superior to US and CT as it provides a higher soft tissue contrast, a more detailed evaluation of the disease’s extent, and the initial site of infection [6,11,13,18,19]. It may be even required if findings from other imaging modalities remain indecisive [11,13]. Detailed imaging, especially at the early stages of the disease with no or sparse symptoms, facilitates diagnosis and adequate surgical debridement, which may improve the outcome, prevent severe complications, and hinder a recurrence [5,9,13].

## 5. Conclusions

Our study underlines the significance of early magnetic resonance imaging in the process of decision-making in cases of suspected Fournier gangrene in order to avoid a potentially fatal course and improve the prognosis.

## Figures and Tables

**Figure 1 diagnostics-12-02320-f001:**
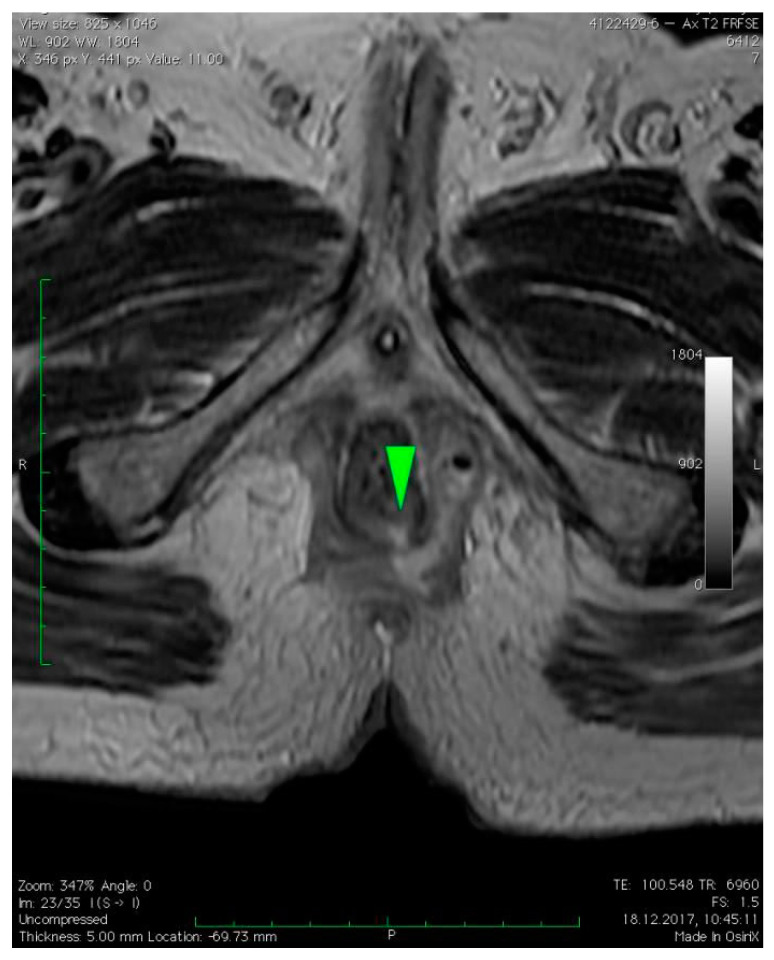
Axial T2–weighted FSE sequence image of perianal fistula on the left side of the posterior anal wall with a width of 5 mm (green arrow) and high signal intensity on T2–weighted FSE sequences, 35 mm from anal sphincter. It connected the anal canal with perianal fluid collection adjacent to the posterior anal wall with dimensions of approximately 15 × 20 × 20 mm. The fluid spread along the left lateral anal wall with a diameter of 10–15 mm.

**Figure 2 diagnostics-12-02320-f002:**
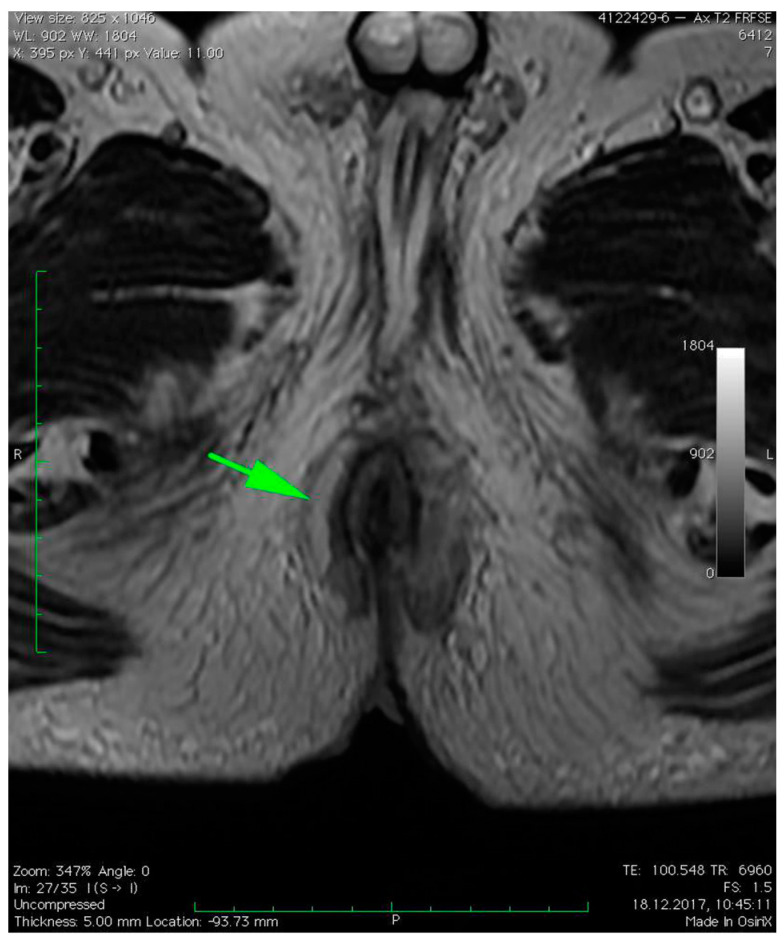
Axial T2–weighted FSE sequence image of right-sided fluid collection spread along the right lateral anal wall with a diameter of 5–10 mm (green arrow).

**Figure 3 diagnostics-12-02320-f003:**
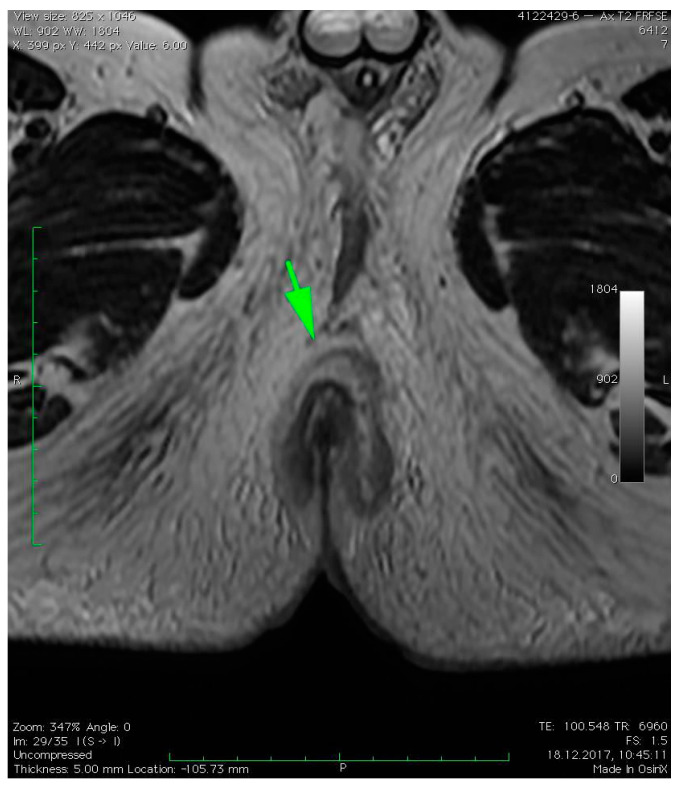
Axial T2–weighted FSE sequence image of fluid collection converging on the anterior wall of the anus (green arrow).

**Figure 4 diagnostics-12-02320-f004:**
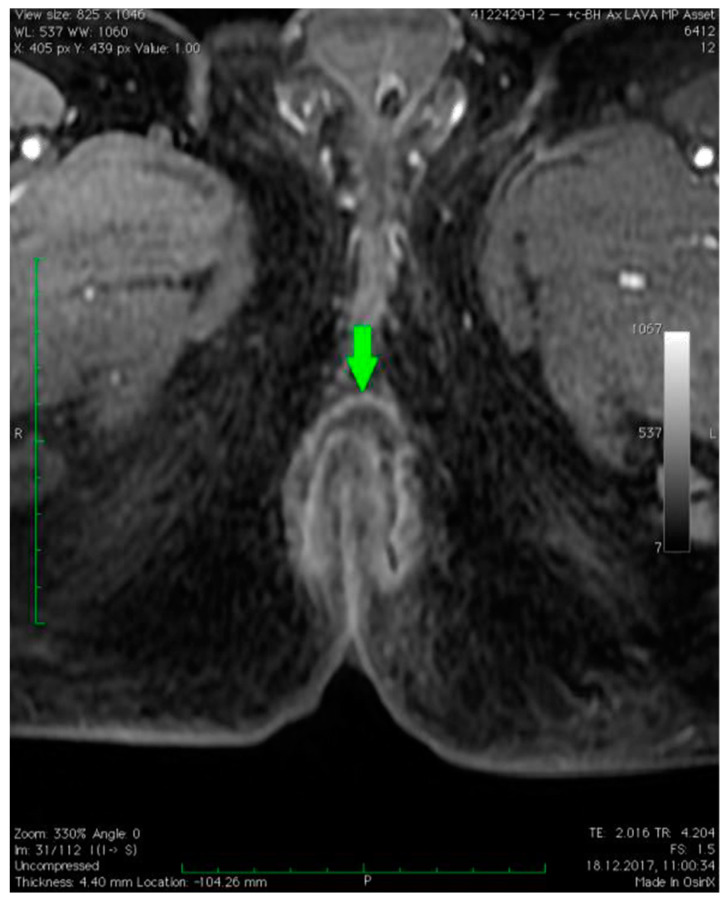
Axial dynamic LAVA gradient echo pulse sequence image of fluid collection connecting the anterior wall of the anus (green arrow).

**Figure 5 diagnostics-12-02320-f005:**
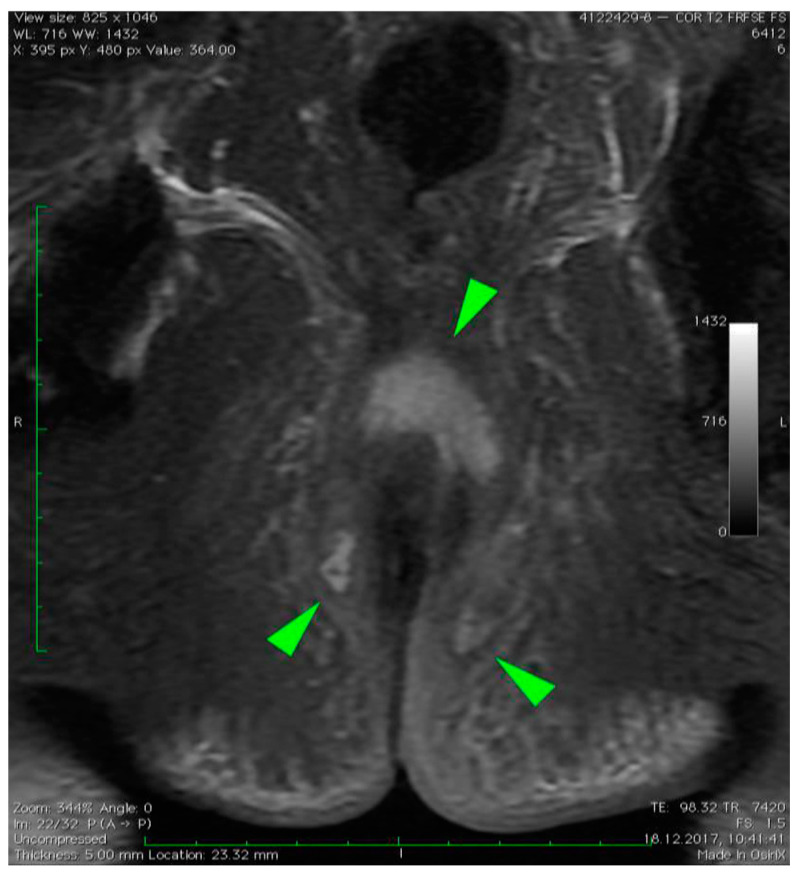
Coronal T2–weighted FRFSE FS sequence image of the perianal abscess with fluid collection on both sides of the anus (green arrows).

**Figure 6 diagnostics-12-02320-f006:**
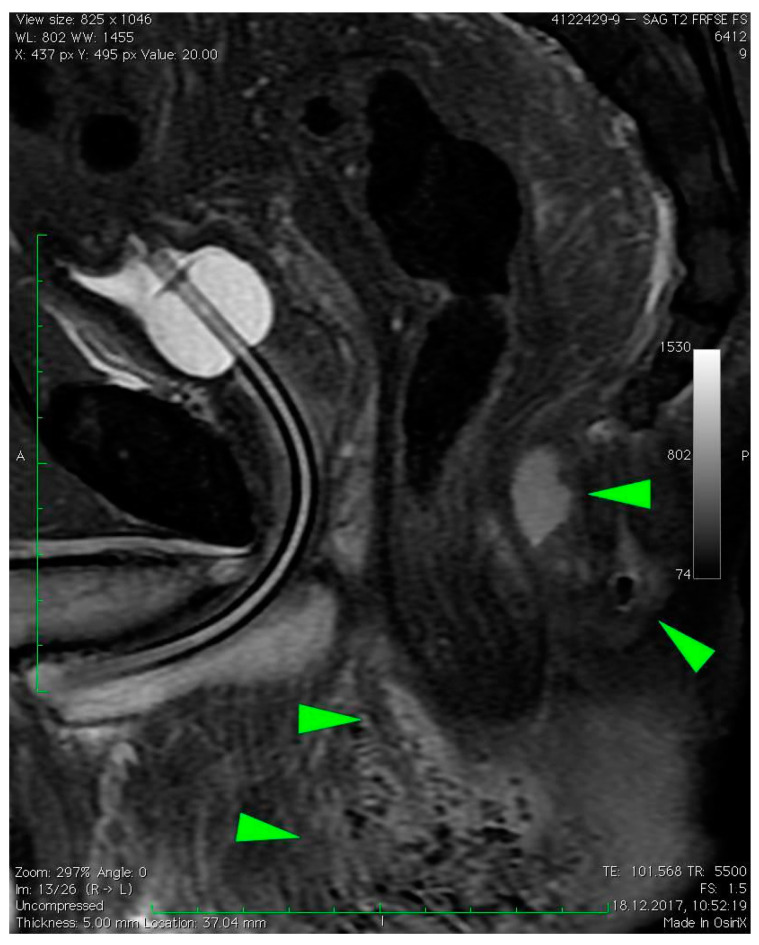
Sagittal T2–weighted FRFSE FS sequence image revealed abnormal signal intensity extending downwards to the buttocks; it also revealed left side inflammatory infiltration with numerous gas foci in the area of 60 mm, which extended to the perineum (green arrows).

**Figure 7 diagnostics-12-02320-f007:**
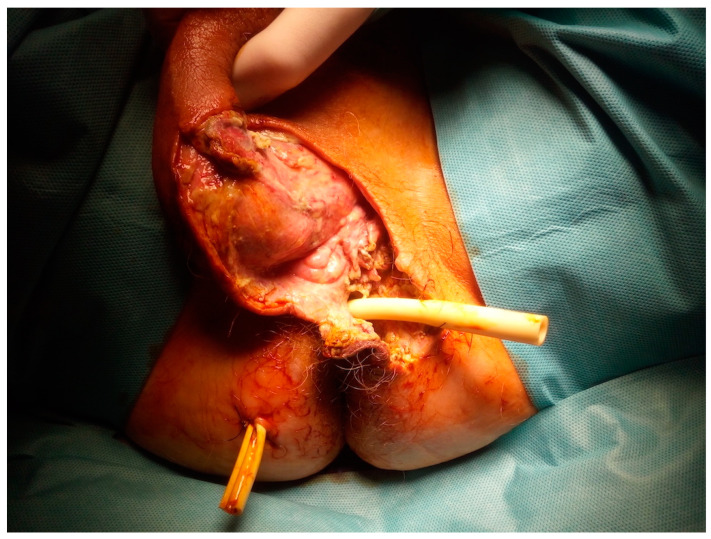
Intraoperative image after scrotal and perianal debridement with left orchidectomy and bilateral drainage of perianal area.

**Figure 8 diagnostics-12-02320-f008:**
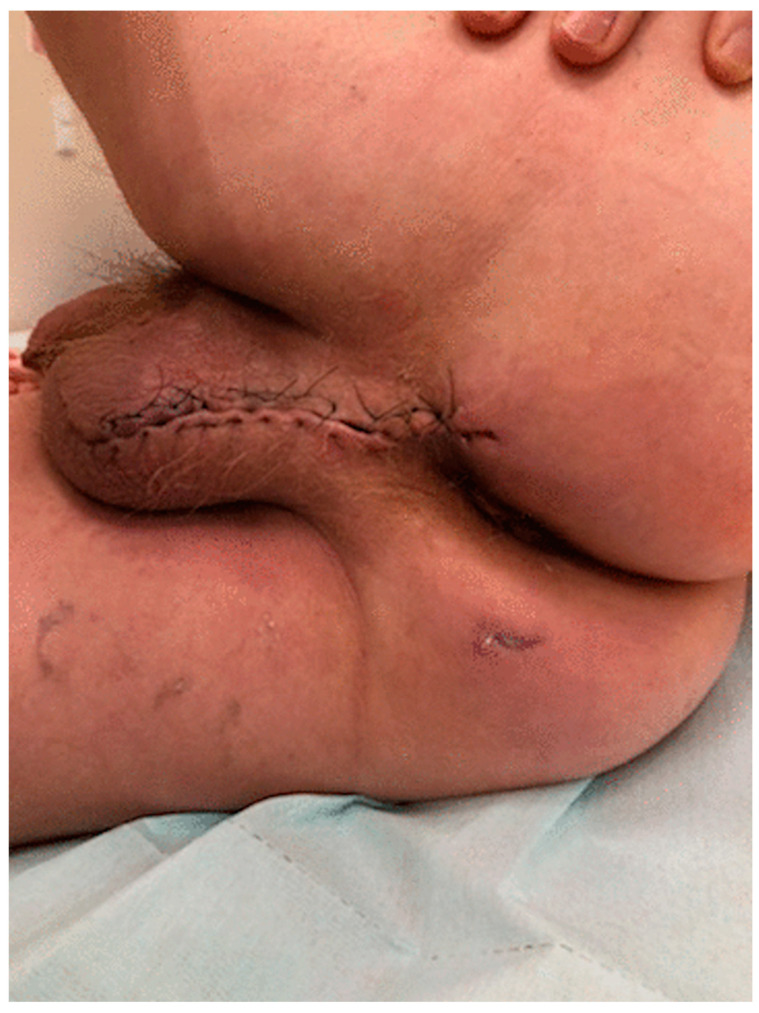
Scrotal and perianal wound after reconstruction.

## Data Availability

Data not contained in the manuscript are available from the corresponding author upon reasonable request.

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
