# Peer review of "Utility of Diagnostic Imaging in the Early Detection and Management of the Fournier Gangrene"

_diagnostics, 2022, doi:10.3390/diagnostics12102320_

Round 1

Reviewer 1 Report

This work is about MODERN DIAGNOSTIC IMAGING IN THE ASSESSMENT OF THE FOURNIER GANGRENE CAUSES. The aim is clear, the title is informative. The study must be modified for english language usage.

References are relevant but they should be enriched. Images are interesting,there are good figures.

Conclusions are supported by references but they should be deepened

Author Response

English language was improved by native speaker.

References along with  conclusions were deepened.

Methods were additionally attached.

Presentation of results was improved.

Corrected or added text was highlighted in yellow.

Reviewer 2 Report

Dear Authors, congratulations for the paper!

I have some suggestions to be added if you consider:

1. Title

Suggest the inclusion of case report to be more clear for the reader, because the title is too broad -MODERN DIAGNOSTIC IMAGING IN THE ASSESSMENT OF THE FOURNIER GANGRENE CAUSES: CASE REPORT

2. Abstract

Should be more descriptive including aims, methods - needs to be clear

3. Introduction

The document I received doesn't have Introduction starts with Fig. 1 Intraoperative image after scrotal and perianal debridement with left orchidectomy and bilateral drainage of perianal area.

I recommend to start with a clear introduction, that you have on the following pages with the description of the pathology, associated complications and  diagnostic technology used - More recent data and references should be included to confirm epidemiological data that corroborate the need to improve early diagnosis and advantages in intervention.

- El-Qushayri AE, Khalaf KM, Dahy A, Mahmoud AR, Benmelouka AY, Ghozy S, Mahmoud MU, Bin-Jumah M, Alkahtani S, Abdel-Daim MM. Fournier's gangrene mortality: A 17-year systematic review and meta-analysis. Int J Infect Dis. 2020 Mar;92:218-225. doi: 10.1016/j.ijid.2019.12.030. Epub 2020 Jan 18. PMID: 31962181.

Lewis GD, Majeed M, Olang CA, Patel A, Gorantla VR, Davis N, Gluschitz S. Fournier's Gangrene Diagnosis and Treatment: A Systematic Review. Cureus. 2021 Oct 21;13(10):e18948. doi: 10.7759/cureus.18948. PMID: 34815897; PMCID: PMC8605831.

The paper starts with an image without any previous reference in the text, as well as the remaining images appear in the sequence of the presentation. The images and video used need to be referenced in the text (names of the images separated from the rest of the test - confuses the reader) 

3. Methods

The Methodology with the presentation of the case study should be separated and described in detail, with the evolution of the decisions as presented in the text, but it must be well defined.

Images are important but need to be organized - the text only as references to fig 1 and 2. The authors included a total of 8 Figures and 1 video.

4. Discussion / Conclusion

After presenting the evolution and findings a Discussion based on other studies should be made, to justify the purpose of the article. 

A conclusion Needs to be included!

I recommend the use of more references to give solidity to the article 

I think the paper is important to increase knowledge on these kind of pathologies and improve the quality of care.

Author Response

1. Title have been rearranged.

2. Abstract was improved.

3.Introduction was added. Presentation of results was improved. More recent data and references were included. The images and video are referenced in text.

3. Methods were attached. Images were reorganised.

4. Discussion was broadened by new aspects. Additional references were added. Conlusions were included.

English language was improved by native speaker.

Corrected or added text was highlighted in yellow.